# Online cognitive–behavioural therapy for traumatically bereaved people: study protocol for a randomised waitlist-controlled trial

Lonneke Lenferink ,[1,2] Jos de Keijser,[1] Maarten Eisma,[1] Geert Smid,[3,4,5] Paul Boelen[2,3,4]

[1]Clinical Psychology and Experimental Psychopathology, University of Groningen, Groningen, The Netherlands
[2]Clinical Psychology, Utrecht University, Utrecht, The Netherlands
[3]ARQ Nationaal Psychotrauma Centre, Diemen, The Netherlands
[4]Foundation Centrum '45, Diemen, The Netherlands
[5]University of Humanistic Studies, Utrecht, The Netherlands

**Correspondence to**
Dr Lonneke Lenferink;
l.i.m.lenferink@rug.nl

## ABSTRACT

**Introduction** The traumatic death of a loved one, such as death due to a traffic accident, can precipitate persistent complex bereavement disorder (PCBD) and comorbid post-traumatic stress disorder (PTSD) and depression. Waitlist-controlled trials have shown that grief-specific cognitive–behavioural therapy (CBT) is an effective treatment for such mental health problems. This is the first study that will examine the effectiveness of online CBT (vs waitlist controls) in a sample exclusively comprised of people bereaved by a traumatic death. Our primary hypothesis is that people allocated to the online CBT condition will show larger reductions in PCBD, PTSD and depression symptom levels at post-treatment than people allocated to a waitlist. We further expect that reductions in symptom levels during treatment are associated with reductions of negative cognitions and avoidance behaviours and the experience of fewer accident-related stressors. Moreover, the effect of the quality of the therapeutic alliance on treatment effects and drop-out rates will be explored.

**Methods and analysis** A two-arm (online CBT vs waiting list) open-label parallel randomised controlled trial will be conducted. Participants will complete questionnaires at pretreatment and 12 and 20 weeks after study enrolment. Eligible for participation are Dutch adults who lost a loved one at least 1 year earlier due to a traffic accident and report clinically relevant levels of PCBD, PTSD and/or depression. Multilevel modelling will be used.

**Ethics and dissemination** Ethics approval has been received by the Medical Ethics Review Board of the University Medical Center Groningen (METc UMCG: M20.252121). This study will provide new insights in the effectiveness of online CBT for traumatically bereaved people. If the treatment is demonstrated to be effective, it will be made publicly accessible. Findings will be disseminated among lay people (eg, through newsletters and media performances), our collaborators (eg, through presentations at support organisations), and clinicians and researchers (eg, through conference presentations and scientific journal articles).

**Trial registration number** NL7497.

## INTRODUCTION

Worldwide, traffic accidents represent the leading cause of unnatural deaths.[1] A total

### Strengths and limitations of this study

► This study is the first to examine the effectiveness of online cognitive–behavioural therapy (CBT) (vs waitlist controls) in reducing psychopathology after traumatic loss in a randomised controlled trial.
► This study is one of the first to examine potential correlates of change in symptom levels following online treatment after traumatic loss.
► We are not able to formally test mediators or moderators of treatment effects.
► We are not able to examine if online CBT has equal effects as face-to-face CBT.
► We are not able to establish formal diagnoses, as we use self-report questionnaires, instead of diagnostic interviews, to assess symptom levels.

of 10%–20% of bereaved people who experience natural deaths (eg, illness) develop severe and persistent grief-related distress, including persistent complex bereavement disorder (PCBD), post-traumatic stress disorder (PTSD) and depression.[2 3] Notably, PCBD has been introduced as other specified trauma-related and stressor-related disorder, in the latest version of the Diagnostic and Statistical manual of Mental Diseases (DSM-5).[4] PCBD can be diagnosed if, after the death of a significant other at least 12 months earlier, a person experiences persistent yearning for the deceased and symptoms of reactive distress (eg, emotional numbness) and social/identity disruption (eg, feeling alone) causing impairment in daily life. While some PCBD symptoms overlap with PTSD (eg, anger) and depression symptoms (eg, diminished interest in activities), several studies have shown that these three syndromes are distinct.[5–7] Unexpected/violent losses of a significant other, also referred to as a traumatic losses, including deaths caused by

traffic accidents, increase risks for the development of PCBD, PTSD and depression.[8 9]

## HEIGHTENED RISK FOR DEVELOPING PSYCHOPATHOLOGY AFTER DEATHS DUE TO TRAFFIC ACCIDENT

Specific circumstances of losses caused by accidents may account for the elevated risk of grief-related distress. For instance, experiencing multiple losses simultaneously, being a witness to the accident, and juridical and financial consequences are proposed to exacerbate grief-related distress.[10] Furthermore, negative cognitions and avoidance behaviours may mediate the influence of sudden/violent loss on grief, PTSD and depression levels.[11] According to a cognitive–behavioural model, three interacting malleable processes underlie disturbed grief reactions: (1) negative cognitions, (2) avoidance behaviour and (3) difficulties integrating the loss into the autobiographical knowledge base.[12]

Experiencing a loss due to a traffic accident may violate basic assumptions about the world being a safe place.[13] This may fuel negative cognitions (eg, 'I'm less worthy, since s/he died' and 'The death of him/her has taught me that the world is unjust') that may exacerbate and maintain acute grief responses.[14] Avoidance behaviours include depressive avoidance and anxious avoidance strategies. Depressive avoidance refers to withdrawal from social and occupational activities that were perceived as fulfilling before the death, out of the conviction that these activities are no longer meaningful. Anxious avoidance strategies serve to prevent confrontation with the reality of the death, out of fear that confrontation is too painful.[12] One potential way to avoid confrontation with the reality of the loss is to focus on angry thoughts and feelings (eg, 'I was angry at the police, courts, or administration, because they did not do their work well enough').[15] This seems to be a frequently used avoidant coping strategy in bereaved people after traffic accidents and is strongly related to PTSD.[16] Difficulty with integration of the loss into the autobiographical knowledge base refers to the difficulties connecting factual knowledge that the loss is irreversible with existing information about the self and the relationship with the lost person, stored in autobiographical memory. Memories related to the loss may lack context in terms of time and place, causing the loss to be experienced as unreal.[17] It has been argued that this 'sense of unrealness' may trigger intrusive memories and increase feelings of numbness or shock once the bereaved person is confronted with reminders of the loss.[17 18] The extent to which a person believes that one is capable of managing stressor-related thoughts, emotions and behaviours, also referred to as self-efficacy (eg, 'I can usually handle whatever comes my way'), has also been determined as an important factor facilitating coping with traumatic stressors.[19] Decreased self-efficacy, negative cognitions and insufficient integration of the loss may contribute to increased sensitivity to loss reminders or secondary stressors following traumatic loss.[20]

## CBT FOR GRIEF-RELATED DISTRESS

Grief-specific CBT has been demonstrated to be the most effective treatment for bereaved people with elevated grief levels.[21–24] CBT targets the abovementioned cognitive-behavioural variables with cognitive restructuring, loss-related exposure and behavioural activation. Notably, research on putative mechanisms of change of grief-specific CBT is sparse[23] (but see refs 25 26). Examining the effectiveness of grief-specific CBT and its potential mechanisms of change in traumatically bereaved people with traumatic grief is clinically relevant because it would enable tailoring of interventions to the specific needs of this group, which could improve treatment outcomes.[27]

While the majority of trials assess the efficacy of face-to-face CBT,[24] so far, to the best of our knowledge, three online CBT-based interventions have been developed for distressed bereaved people.[28–30] These prior studies provided preliminary data on the potential effectiveness of online grief-specific CBT, but had some limitations. For instance, treatment was solely provided to people who experienced perinatal loss[29] or included relatively small samples.[28] Comparability between these three studies is also limited, because interventions differed in treatment content; different elements of CBT were offered, for instance, behavioural activation, exposure[28] or writing assignments.[29 30] Offering CBT via the internet has some potential advantages. It may lower the threshold for seeking treatment, because it can be delivered independent of geographical location. Furthermore, asynchronous communication may be used, allowing the client and therapist can contact each other at any preferred time.[31] This may counter barriers to mental health service use, such as difficulties with finding help, transportation concerns or difficulties scheduling treatment sessions.[32] In addition, online CBT could reduce treatment costs, improving accessibility and dissemination of care for people in need of support.[33] Moreover, during times of a crisis, such as the COVID-19 pandemic, it seems more relevant than ever to further examine the effectiveness of online CBT for distressed bereaved people, as it will allow them to retain access to evidence-based care.[34]

A potential downside to online CBT is the high dropout rate found in earlier studies.[33 35] It has been argued that a strong therapeutic alliance might support adherence to online treatment and mediates treatment effects.[36] Therapeutic alliance is defined as a positive emotional bond between client and therapist, whereby both parties agree on the tasks and goals of the treatment.[37] The client–therapist relationship might also explain why online treatments are more effective with therapist guidance than without.[31] Concerns have been raised that developing a therapeutic relationship might be more difficult when non-verbal communication is absent.[38] However, studies in non-bereaved samples indicate that developing

a strong therapeutic alliance is possible during online treatment[33] and that therapeutic alliance is often related to online treatment outcomes,[39] but not always.[33] More research is needed to further examine the interrelations of the quality of client–therapist relationship, drop-out and treatment outcomes in online CBT.

## STUDY OBJECTIVES

Our first aim is to examine the effectiveness of online CBT (vs a waiting list control condition) in reducing symptom levels of PCBD, PTSD, and depression in people bereaved by a traffic accident. We expect that participants assigned to the online CBT condition will show larger reductions in symptom levels of PCBD, PTSD and depression compared with waitlist controls at post-treatment assessments (hypothesis 1).

Our second aim is to explore correlates of change. Based on prior research and theories,[12 16 19] we expect that reductions in negative cognitions, avoidance behaviours, state anger, a sense of unrealness and improvement in self-efficacy are related to reductions in PCBD, PTSD and depression levels in online CBT (hypothesis 2a). Additionally, we aim to explore whether background characteristics (ie, gender, age and educational level, kinship to the deceased and time since loss) and accident-related stressors (ie, single vs multiple loss, witnessing the accident and status of legal trial) are related to treatment effects (hypothesis 2b). We have no specific expectations regarding these associations because prior treatment studies in bereaved people showed inconsistent results.[24 25 40] However, based on clinical experience, we expect that accident-related stressors are associated with treatment effects, such that multiple loss, witnessing the accident and ongoing legal trial negatively impact treatment effects.

Our third aim is to explore the associations between quality of the therapeutic alliance and drop-out rates and treatment outcomes. We expect that a stronger therapeutic alliance is related to lower drop-out rates and better treatment outcomes.

## METHODS AND ANALYSIS
### Design

A two-arm (online CBT vs waiting list) multicentre open label parallel randomised controlled trial (RCT) will be conducted. Randomisation will take place after the participant is screened for eligibility-based inclusion criteria (described below). A random number generator (www.random.org) will be used by a blinded independent researcher, to perform the blocking randomisation procedure. An allocation ratio of 1:1 will be applied.

Participants allocated to the online CBT condition receive treatment within 1 week after allocation. All participants will be asked to fill in questionnaires (described below) at baseline (T1), 12 weeks postallocation (T2 for the intervention condition and T1a for waitlist controls) and 20 weeks postallocation (T3 for the intervention condition and T1b for waitlist controls). For participants in the waiting list control group, at the end of the 20-week waiting period after which they will receive online CBT, they will be asked to fill in T2 and T3 12 and 20 weeks after starting treatment, respectively (see figure 1). A link to online questionnaires will be sent to the participants by a non-blinded member of the research team at each time point. A waitlist control group (instead of a no treatment control group) is chosen to increase the likelihood of continued study participation by guaranteeing that all participants receive treatment. Furthermore, the inclusion of a waiting list control group allows a treatment versus no treatment comparison that will provide knowledge about the effects of treatment relative to natural recovery from loss.

In line with prior treatment studies from our research group,[40 41] the online treatment is guided by governmentally licensed psychologists, connected with a Dutch informal 'traumatic loss network' of therapists specialised in treating emotional distress following traumatic loss. In total six therapists (including authors PB and JdK who are registered clinical psychologists) will guide the participants online; participants will receive feedback from the same therapist each time. The therapists will receive a training, provided by LL, PB and JdK, on

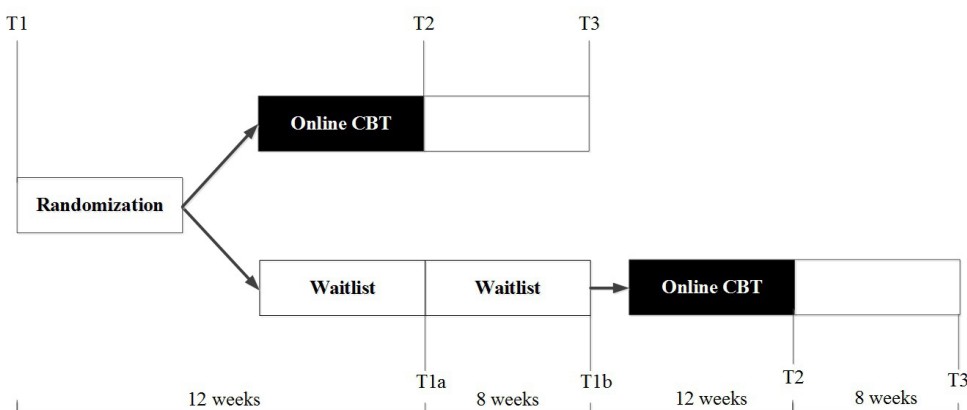

**Figure 1** Design of RCT. CBT, cognitive–behavioural therapy; RCT, randomised controlled trial.

the use of the treatment protocol of this intervention study. In preparation for the training, therapists read all treatment materials and a selection of grief treatment literature. Instructions about the use of the online treatment interface will be given by its developers. During a 5-hour face-to-face group meeting the rationale of the online treatment will be explained and research procedures will be discussed. In a 2-hour online video meeting outstanding questions regarding the treatment and the research project will be answered. Supervision (by telephone or email) by PB and JdK is possible on request, for instance, when therapists encounter difficulties in treatment. Therapists will be contacted by a member of the research team by phone or email biweekly to monitor treatment progress and protocol adherence. Treatment costs will be reimbursed.

## Participants

This RCT is part of a larger ongoing research project (the 'TrafVic-project') examining the psychological impact of, and care after, the death of a loved one due to a traffic accident. We expect to recruit the majority of the participants via a survey that started in December 2018 and included the following question: 'In this study we would like to offer psychological help to persons who experience emotional problems. May we approach you with more information about this offer, if your answers to this questionnaire show that you experience emotional problems?' Those who answered 'yes' will be sent a letter with information about the intervention, the treatment study and an informed consent form (see online supplementary file). A Dutch website (www.rouwnaverkeersongeval. nl) has been developed so that potential participants can read information about the research and treatment. People who are interested can also sign up for the study via this website. Recruitment for this RCT had not started at the time of submission of this manuscript.

To be eligible for study participation, the person must (1) be a family member, spouse or friend of a person who died due to a traffic accident at least 1 year earlier, (2) be ≥18 years of age and (3) meet DSM-5 criteria for PCBD and/or PTSD and/or experience clinically relevant depression, based on questionnaire scores (see below for more details). People are excluded when they do not master the Dutch language or have no internet access.

## Sample size

To test our primary hypothesis (hypothesis 1), a test for each outcome separately (PCBD, PTSD and depression) will be conducted to assess the effects of online CBT versus waitlist controls. To find a difference between two groups (online CBT vs waitlist controls) of at least a medium effect size (f=0.25; based on prior research[22 28 40]) with a power of 80%, an α of 0.017 (corrected for multiple testing, that is, 0.05/3, as there are three primary outcome measures (PCBD, PTSD and depression), and a strong association (r=0.50) between the preassessment and postassessment, a sample size of 23 per condition is sufficient. Taking into account an average dropout rate of 19%,[22] a total sample size of 55 (46+9) is required to test hypothesis 1.

Because our data are nested (repeated measures) (level 1) within individuals (level 2), and possibly within families sharing the same household (level 3), multi-evel modelling will be performed to test hypothesis 1. Conducting a power analysis within a multilevel framework is not feasible for various reasons.[42] Our power analysis is therefore based on a repeated measures analysis of variance.

## Intervention

Online CBT will consist of eight one-on-one sessions, called lessons, offered within a timeframe of 12 weeks. Eight sessions have shown to be sufficient to yield clinically relevant effects in prior research.[40] Following Dutch guidelines for grief-specific CBT,[42] central components of the treatment are exposure, cognitive restructuring and behavioural activation. In the first session, psychoeducation is offered, including information about possible emotional reactions to the death of a loved one in a traffic accident and processes that might foster or hamper recovery. A rationale for the CBT interventions is provided.

Then, sessions 2–4 are focused on exposure; the circumstances and story of the loss are presented in detail, and the participant is encouraged to confront stimuli that s/he tends to avoid. Exposure is conducted by imaginary exposure assignments and by writing assignment that have proven to be effective in prior research.[30] These writing assignments are focused on writing a detailed narrative of the loss and its circumstances.

The next sessions (5 and 6) focus on identifying and changing negative cognitions that hamper adjustment (ie, cognitive restructuring); specific attention is paid to cognitions connected with responsibility/guilt and anger that may be experienced following the accidental death.[10] Cognitive restructuring assignments are provided to gain an alternative perspective on negative thoughts about the self, life, the future, through (1) psychoeducation about common unhelpful thoughts, (2) identifying one's own unhelpful thoughts and (3) challenging these thoughts. Participants are instructed to undertake these three steps by providing a daily description of (1) an emotional moment/event, (2) their thoughts during this event, (3) their feelings (and intensity of these feelings on a scale of 1–10), (4) their behaviour, (5) evaluation of their thoughts) and (6) alternative helpful thoughts.

In sessions 7 and 8, participants are encouraged to re-engage in previously valued social, recreational and occupational activities in order to facilitate the process of adjustment. Behavioural activation assignments are focused on writing about valued activities and making plans to achieve valued goals. Session 8 is also focused on what the participant has learnt and how to deal with difficulties in the future.

All information and assignments are presented in an online framework, offered via a secure website. Participants receive online written information that consists of

psychoeducation, information about treatment content and structure, and homework assignments. As part of the online treatment, participants also listen to a video therapist verbally sharing parts of information that are also presented in text. The video therapists are two members from the traumatic loss network; one male and one female psychotherapist who are middle aged and specialised in treating bereaved people. At the start of the treatment the video therapists introduce themselves and the participant is asked to select one of the video-therapists. The information shared by these video therapists are recorded in video messages in which they read parts of the texts out loud. Each participant, therefore, receives the same information from a video therapist. Direct contact with the video therapist is not possible.

Participants receive weekly asynchronous written feedback from one online therapist on each assignment that they complete online. As mentioned earlier, six online therapists are trained to guide the participants. The online therapists are instructed to contact the participant twice a week; once to encourage participants to log in and complete assignments and once to provide feedback on assignments. In total, they spend 30 min per week on reading assignments and providing feedback. Moreover, participants are encouraged to ask a family member or friend to support them during treatment. This support figure is then informed about the treatment through written information in an online framework.

## Measures
### Primary outcome measures
PCBD will be assessed with the Traumatic Grief Inventory-Self Report (TGI-SR).[43] The TGI-SR consists of 18 items on a 5-point Likert scale ranging from 1=never to 5=always. Four items tapping disturbed grief criteria according to the 11th edition of the International Classification of Diseases were added.[44] An example of an item is: 'I found it difficult to trust others'. The instruction of the original questionnaire was altered from referring to 'the death of your loved one' to 'the death of your loved one(s) due to a traffic accident'. Psychometric properties of the TGI-SR are adequate.[43 45] Participants are considered to meet criteria for DSM-5 PCBD[4] when they score at least 3 ('sometimes') on at least 1 criterion B symptom (Item 1, item 2, item 3 and item 14), at least six criteria C symptoms (item 4 up to 11, and item 15 up to 18) and the criterion D symptom (item 13).

PTSD will be assessed with the PTSD checklist for DSM-5 (PCL-5)[46] (Dutch version[47]). Participants rate how often they were bothered by each symptom (eg, 'In the past month, how much were you bothered by trouble remembering important parts of the accident?') on 5-point Likert scales (0=not at all and 4=extremely). The instruction and the items of the original questionnaire are altered from referring to the 'stressful event' to the 'the death of your loved one(s) due to a traffic accident'. The PCL-5 has shown to be reliable and valid.[46] Participants meet the criteria for DSM-5 PTSD[4] when they score

at least 2 ('Moderately') on 1 criterion B item (items 1–5), 1 criterion C item (items 6–7), 2 criterion D items (items 8–14), and 2 criterion E items (items 15–20).

Depression symptom levels are assessed with the depression subscale of the HADS-D.[48] The HADS-D consists of seven items (eg, 'I still enjoy the thing I used to do') rated on 4-point scores ranging from 0 (eg, 'Hardly at all') through 3 (eg, 'Definitely as much'). The Dutch HADS-D is a reliable and valid screening tool for depression.[49] A cut-off score of ≥8 is used as indicator for clinically relevant depression.[48]

### Secondary outcome measures
Negative grief-related cognitions are assessed with 18 items from the Grief Cognitions Questionnaire.[14] Participants are asked to rate their agreement with each item (eg, 'Since [–] is dead, I feel less worthy') on 6-point scales varying from 0=disagree strongly through 5=agree strongly. The psychometric properties have been positively evaluated in prior research.[14]

Avoidance is measured with the Depressive and Anxious Avoidance in Prolonged Grief Questionnaire (DAAPGQ).[50] The depressive avoidance subscale consists of 5 items (eg, 'Since [–] is dead, I do much less of the things that I used to enjoy.') and the anxious avoidance subscale consists of 4 items (eg, 'I avoid to dwell on painful thoughts and memories connected to his/her death.'). Participants answer each item on an 8-point scale with 0=not at all true for me, and 7=completely true for me. The DAAPGQ has adequate psychometric properties.[50]

State anger is assessed with the 15-item state anger subscale of the State-Trait Anger Expression Inventory-2 (STAXI-2)[51] (Dutch version:[52]). Participants are asked to rate on 4-point Likert scales (1=not at all and 4=extremely) how angry they feel right now (eg, 'I feel annoyed'). The STAXI-2 is a valid and reliable measure to assess state anger.[52]

A sense of unrealness is measured with the 5-item Experienced Unrealness Scale.[17] Participants are asked to rate their agreement with each item (eg, 'I still can hardly imagine that [–] will never be here again') on 8-point scales (0=not at all true for me 7=completely true for me). This instrument demonstrated adequate psychometric properties.[17]

Self-efficacy is assessed with the General Self-Efficacy Scale (GSES).[53] The GSES is a 10-item measure. Participants are asked to rate their agreement with each item (eg, 'I can solve most problems if I invest the necessary effort.') on a 4-point scale (1=completely not true, 4=completely true). The GSES has shown excellent reliability and validity.[53]

Quality of the therapeutic alliance is measured with the 12-item Work Alliance Inventory-Short Form, Client Version and Therapist Version after session 4 (WAI-SF)[54] (Dutch version[55]) The WAI-SF consists of 12 items (eg, Client version: 'We agree on what is important for me to work on', Therapist Version: 'We are working towards mutually agreed on goals.') that are rated on 5-point

scales (1=never and 5=always). Higher total scores indicate a higher quality of the therapeutic alliance as perceived by the participant and therapist. The WAI-SF is a reliable and valid assessment tool.[56]

## Other measures

Background characteristics (ie, gender, age and educational level, kinship to the deceased and time since loss) and accident-related stressors (ie, single vs multiple loss, witnessed the accident and status of legal trial) will be assessed with single items.

Participants are allowed to receive other forms of psychosocial, instrumental or legal support during participation in the trial. Using a single question, we will assess whether the participants received other forms of psychosocial professional support. The following question will be used: 'During the past 12 weeks/8 weeks (for T2 and T3, respectively) did you receive additional psychological professional support from a psychologist, therapist or psychiatrist other than the (online) therapist from the TrafVic-study?' We will also include two dichotomous items (yes/no) at T1 to assess psychological support received prior to participation in the study, namely: 'Did you ever receive support from a psychologist, therapist or psychiatrist, for your own emotional/mental problems, prior to the loss of your loved one due to a traffic accident?' and 'Did you ever receive support from a psychologist, therapist or psychiatrist, for your own emotional/ mental problems, related to the loss of your loved one due to a traffic accident?'

## Statistical analyses

To examine the differences in reductions of symptom levels of PCBD, PTSD and depression from pretreatment to post-treatment/waiting period between the conditions (online CBT vs waitlist), three independent multilevel models will be built (hypothesis 1). Symptom levels of PCBD, PTSD, and depression will consecutively be included as dependent variables and condition (online CBT vs waitlist controls), time and time × condition (interaction term) as predictor variables, taking into account that repeated observations (level 1) are nested within individuals (level 2), and within households (level 3; if applicable). Additionally, relevant background, loss-related variables and use of cointerventions (yes/no) during participation in our study will be included in the analysis as covariates. Deviance tests will be used to examine whether inclusion of these covariates improves model fit.[57] Data of all participants entering the study will be included in all analyses (ie, intention-to-treat analysis). Furthermore, percentages of people meeting diagnostic criteria for PCBD, PTSD and clinically relevant depression will be calculated for each measurement occasion and percentages of people reporting reliable change scores for each outcome measure, using a formula from Jacobson and Truax,[58] p 14 will be reported.

To examine to what extent symptom improvement after treatment is related to improvement in possible correlates of change, residual gain scores will be calculated for all outcome measures (ie, PCBD, PTSD and depression) and possible correlates of change (ie, negative cognitions, avoidance behaviours, state anger, a sense of unrealness and self-efficacy). Following previous research[59], residual gain scores will be calculated by subtracting the standardised combined pretreatment scores of both conditions (T1 data from immediate treatment condition and T1b data from waitlist condition) multiplied by the correlation coefficient between standardised combined pretreatment scores and standardised post-treatment (or follow-up) scores from standardised post-treatment (or follow-up) scores. To test hypothesis 2a, multiple regression analyses will be conducted to examine the associations between residual gain scores of PCBD, PTSD or depression and residual gain scores of negative cognitions, avoidance behaviours, state anger, a sense of unrealness and self-efficacy.

To achieve research aim 2b, multiple regression analyses will be used to examine to what extent residual gain scores of PCBD, PTSD and depression varies as function of a) background characteristics, including gender (male/female), age (in years) and educational level (low/ high), kinship to the deceased (child/spouse vs other) and time since loss (in years) and b) accident-related stressors, including number of losses (single vs multiple), witnessing the accident (yes/no) and status of legal trial (not applicable/ongoing/completed). Condition (intervention vs waitlist controls) will be added as a covariate to fulfil research aims 2a and 2b.

To achieve the third research aim: (1) differences in therapeutic alliance scores will be assessed between people who completed and dropped out of treatment and (2) multiple regression analyses will be used to examine to what extent symptom improvement in PCBD, PTSD and depression is related to therapeutic alliance (from both participant and therapist perspectives).

## Ethics and dissemination

The initial plan for this study was to conduct a three-arm (face-to-face CBT, online CBT and waiting list) RCT to examine the effectiveness of face-to-face CBT (vs waitlist controls) and online CBT (vs waitlist controls). This study has been approved by the Medical Ethics Review Board of the University Medical Center Groningen (METc UMCG: ID number: M20.252121). Due to the COVID-19 outbreak, we had to change our study protocol, because face-to-face contact with a therapist was not possible because of social distancing measures. Instead of comparing the effects of online and face-to-face CBT with waitlist controls, we changed the design of the study by comparing the effects of online CBT versus waitlist controls before enrolment of participants took place. This amendment to our study has been approved by the same ethics committee.

The study will be conducted according to the principles of the Declaration of Helsinki (eighth version, 2013) and in accordance with the Medical Research Involving Human Subjects Act. Collected data will be handled

confidentially, according to the European Union General Data Protection Regulation and the Dutch Act on Implementation of the General Data Protection Regulation. Unidentifiable data from this trial will be stored in data repositories from the University of Groningen and Utrecht University.

Findings of this RCT will be disseminated among participants by means of a newsletter. If shown to be effective, the online framework will be made publicly accessible, so that it can benefit other bereaved people. Findings will also be disseminated among lay people by uploading the newsletters on our website (www.rouwnaverkeersongeval.nl) and through media performances. Our findings will be presented to our collaborators, including non-governmental organisations and (peer-)support organisations for bereaved people. Treatment materials will also be made available on request. Lastly, colleagues will be informed about our findings during presentations at (inter)national conferences and publications in scientific journals.

## Patient and public involvement

At the start of this project an advisory committee was established. This committee includes someone who lost a significant other after a traffic accident, a lawyer with expertise in supporting bereaved people after traffic accidents, and representatives of Victim Support the Netherlands and Fund Victim Support. This committee was involved in the development of the research questions, outcomes measures and design of the study by reading and commenting on drafts of our research proposal and study protocol. This committee pilot tested the questionnaires and was involved in the development of recruitment materials, recruitment strategies, and information materials for participants by reading, revising and approving the drafts. This committee helps the research team in recruiting participants by sharing information about this study in their own professional network. The advisory committee is not involved in conducting the study or development of treatment materials. The committee will support the research team when disseminating the study findings among relevant audiences by help writing and reviewing newsletters and press releases.

## DISCUSSION

The relatively few RCTs among general bereaved people with elevated grief levels indicate that grief-specific CBT-based interventions yield the largest effects on postloss mental health compared with a waiting list.[21–24] RCTs evaluating face-to-face or online treatment effects for people with elevated mental health complaints after confrontation with sudden/violent losses are lacking, with the exception of two studies that compared face-to-face EMDR plus CBT against waitlist controls.[40 59] Given that traumatically bereaved people are at risk for PCBD and comorbid PTSD and depression,[8] it seems particularly relevant to develop evidence-based interventions for this population.

This will be the first RCT to examine the effectiveness of online CBT in a sample exclusively comprised of people who experienced a traumatic death. We are not able to test whether the online CBT has equal effects as face-to-face CBT. Nonetheless, the findings are expected to yield important insights in the effects of online CBT. In this RCT, the online treatment is designed to be as similar as possible to face-face CBT in terms of treatment content, treatment duration, and experience and training of therapists. When we find effect sizes for online CBT that are similar to effect sizes found in earlier studies for face-to-face CBT, delivering CBT online can be considered as supplement to face-to-face treatment, in particular when barriers to face-to-face treatments, such as waiting lists and travel expenses, are experienced.

We will also examine potential correlates of change. These analyses, examining the associations between reductions in symptoms levels and among others negative cognitions and avoidance behaviours, will provide insights in potential underlying therapeutic processes to foster recovery from traumatic loss. These insights are deemed important to design treatments that more effectively target these correlates of change. We also expect to improve our knowledge on for whom (eg, women or people who are more remotely bereaved) grief-specific CBT works best. Findings on these potential correlates of change are necessary to improve treatments given that a maximum of 42% of bereaved people report clinically relevant reductions in grief levels after treatment.[21]

Lastly, the role of therapeutic alliance on therapy outcomes will be explored. Prior research in bereaved people has shown that greater therapeutic alliance, from the perspective of the client, at week 4 of a face-to-face grief-specific treatment, was related to greater reductions in grief levels. This therapeutic alliance–grief relationship was not significant for a non-grief-specific treatment.[60] Our exploration of this association, from the perspective of client and therapist, may for the first time shed light on therapeutic processes in online CBT for traumatic grief.

An anticipated limitation of our RCT is the self-selected sample. It is possible that people who are more open towards innovative technology in general[61] and who received support prior to the loss[32] are more likely to sign up for this study, limiting the generalisability of findings emerging from this study. Due to the absence of an active control group (eg, face-to-face CBT), we are not able to test the effects of online CBT compared with an alternative treatment. Furthermore, we will use self-report measures instead of diagnostic interviews, which may increase the risk of overestimating symptom levels.[62] In addition, participants might experience difficulties with completing the mid-treatment assessment of the therapeutic relationship because the video therapist that provides information through recorded video messages (interaction between video therapist and participant is not possible) might be a different person than the online

therapist who provides personal written feedback twice a week. Although the instructions of the therapeutic alliance measure explicitly refer to the interaction with the online therapist (not the video therapist), this might still be confusing for some participants. Another potential limitation of this trial relates to the fact that the operationalisation and assessment of grief as a disorder is still under debate.[63–65] For instance, PCBD, included as 'condition for further study' in the DSM-5, is likely to be changed in a revision of the DSM. To maximise diagnostic compatibility, we added four items to the TGI-SR, corresponding to Prolonged Grief Disorder (PGD) criteria according to the 11th edition of the International Classification of Diseases (ICD-11), enabling operationalision of our primary outcome measure in terms of diagnoses of pathological grief according to both the DSM-5 and the ICD-11.

To conclude, this RCT will provide new insights in effectiveness of online CBT for people who experience clinically relevant distress after bereavement due to a traffic accident, as well as in potential correlates of therapeutic change. As trials to date have primarily focused on effects of face-to-face treatment for non-traumatically bereaved people, our findings are expected to provide a valuable addition to the knowledge base on treating severely distressed bereaved people.

**Contributors** JdK is principal investigator. LL is executive researcher. JdK, PB, ME and GS are grant holders. LL developed the study design and wrote the ethics proposal and drafts of the manuscript. JdK, ME, GS and PB read, revised, and approved the drafts of the study design, ethics proposal and the manuscript.

**Funding** Fund Victim Support subsidized this work (grant number: not applicable).

**Competing interests** None declared.

**Patient and public involvement** Patients and/or the public were involved in the design, or conduct, or reporting, or dissemination plans of this research. Refer to the Methods section for further details.

**Patient consent for publication** Not required.

**Provenance and peer review** Not commissioned; externally peer reviewed.

**ORCID iD**
Lonneke Lenferink http://orcid.org/0000-0003-1329-6413

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
