## [Reviewer comments · BMJ Open]

ARTICLE DETAILS

TITLE (PROVISIONAL)	Online cognitive behavioral therapy for traumatically bereaved people: Study protocol for a randomized waitlist-controlled trial
AUTHORS	Lenferink, Lonneke; de Keijser, Jos; Eisma, Maarten; Smid, Geert; Boelen, Paul

VERSION 1 – REVIEW

REVIEWER	Dr Filipa Alves-Costa Barnet Enfield and Haringey Mental Health NHS Trust, Forensic Service
REVIEW RETURNED	06-Feb-2020

GENERAL COMMENTS	The protocol submitted has high standards regarding previous literature, data collection and analytical plans, which are likely to address your research aims. Limited research has been conducted to date, thus this study is crucial. Thinking about this protocol and future manuscript, I would like to highlight some recommendations: 1. Abstract: this can benefit from clarity by adding the following headings: Background, Objectives/Aims, Participants and settings, Methods, Results, Conclusions.2. Introduction: This can be more succinct; topics can be better integrated.3. Variables to consider (pg 8). You have considered key variables which will be controlled using adequate statistical analysis. I wonder if the following variables have been considered? Previous literature exploring traumatic bereavements (homicide) have highlighted them. They might contribute to the development of psychopathology; sustained mental health difficulties.- demographic characteristics of both victims and participants- criminal/judicial processes (finished? on-going? "satisfactory outcomes?", positive experience when in contact system?)- Media intrusion4. It would be also important to clarify:- any other current or past Psychology interventions- and/or medical/pharmacological treatment5. Pg 11 - Can you clarify if the research team is going to be (or not) involved in psychology provision?6. Pg 12 - Can you provide more detail about the clinical judgment clinicians will have to make in order to following your inclusion and
---

	exclusion criteria. Is this going to be discussed and agreed to reduce bias? If so, how? 7. Pg - Can you clarify what training and supervision therapists will receive? 8. Will trauma-informed CBT be considered? 9. What safeguarding arrangements do you have in place? 10: Can you clarify if sessions will be 1:1 or group setting? Overall, manuscript reads well, however some paragraphs are too lengthy and lack of clarity.
--	---

REVIEWER	KEE-HONG CHOI KOREA UNIVERSITY, KOREA
REVIEW RETURNED	13-Apr-2020

GENERAL COMMENTS	The current study protocol proposed to examine the efficacy of face-to-face CBT and online CBT as compared to waitlist control group in reductions of PCBD, PTSD, and depressive symptoms. In addition, the authors proposed to explore potential moderators and mediators of change in those symptoms. The authors proposed two important but distinct study aims: (1) efficacy of face-to-face and online CBT, and (2) treatment mechanisms of CBT. Even though two study aims are important and interesting, the mixture of the two aims would cause concerns especially about sample size. The authors said that due to a small sample, the online CBT is not compared to face-to-face CBT even though online CBT for PCBD is quite novel (and having distinct treatment components as indicated by the authors) and unknown for its efficacy. In addition, since investigation of treatment mechanisms requires a large number of subjects, it is unclear why the authors include the online CBT (without examining its efficacy) in the study design for exploring treatment mechanisms. For the treatment mechanisms, it would be better increasing sample size of the face-to-face CBT. It would be helpful for the readers if the authors commented on the effects of moderators in other related studies (though with a different population), such as the fact that male, younger age, and higher level of education were associated with outcomes in PTSD studies (Ehlers et al., 2013). Moreover, I encourage t the authors to address their hypothesis regarding the background characteristics based on clinical experience AND the results from other related studies. In the manuscript, only the hypothesis concerning accident-related stressors was noted. How to control for the practice effects on repeated measures of waitlist controls after allocating to either face to face or online CBT? What if participants who are allocated to waitlist control become no longer eligible for the study due to their natural recovery after 20 weeks? Please elaborate on the qualifications of the personalized therapists for online sessions. For example, are they licensed and registered psychologists who will also receive trainings that are qualitatively equivalent to the 8-hour training that the face-face
--

	CBT therapists would receive? (it was somewhat unclear in the manuscript). Please provide more information on the video-therapist that will conduct the online lessons. Who will be the person conducting the online-lessons (e.g., licensed psychologist, gender, age, etc)? Will the instructor of the online-lessons be the same as the one who will be providing weekly asynchronous feedbacks? This seems like an important issue as the third hypothesis of the study examines the extent to which therapeutic alliance differ in each condition. Will there be a specific guideline for providing weekly asynchronous feedbacks? For instance, will the therapists be encouraged to provide feedbacks within a certain time period such as 2 or 3 days? Who will administer measures, and how to confirm that the measures are administered blind to a study design? Would therapists be blind to a study design? More detailed information about the differences in therapeutic contact (e.g., different doses of guidance, different forms of communication mode) between two therapeutic arms should be helpful.
--	---

VERSION 1 – AUTHOR RESPONSE

Reviewer: 1

Reviewer Name
Dr Filipa Alves-Costa

Institution and Country
Barnet, Enfield and Haringey Mental Health Trust - Forensic Psychologist
Honorary contracts
University of Bath - Lecturer (Clinical)
IoPPN - Research Fellow
United Kingdom

Please state any competing interests or state 'None declared':
None declared.

Please leave your comments for the authors below
Dear authors,

The protocol submitted has high standards regarding previous literature, data collection and analytical plans, which are likely to address your research aims. Limited research has been conducted to date, thus this study is crucial.

Response: Thank you dr. Filipa Alves-Costa for your positive words.

Thinking about this protocol and future manuscript, I would like to highlight some recommendations:

1. Abstract: this can benefit from clarity by adding the following headings: Background, Objectives/Aims, Participants and settings, Methods, Results, Conclusions.

Response: We followed the author guidelines from BMJ Open regarding structure of the abstract for protocols. We are reluctant to deviate from these guidelines and therefore did not change the structure of the abstract.

2. Introduction: This can be more succinct; topics can be better integrated.

Response: We rephrased wording in our Introduction section with the intention to be more concise. We also removed text that provided more information on face-to-face CBT (due to removing the face-to-face CBT condition from our design). As a result, our Introduction section is now less lengthy.

3: Variables to consider (pg 8). You have considered key variables which will be controlled using adequate statistical analysis. I wonder if the following variables have been considered? Previous literature exploring traumatic bereavements (homicide) have highlighted them. They might contribute to the development of psychopathology; sustained mental health difficulties.

- demographic characteristics of both victims and participants
- criminal/judicial processes (finished? on-going? "satisfactory outcomes?", positive experience when in contact system?)
- Media intrusion

Response: Thank you for pointing this out. We do plan to examine the associations between changes in symptom levels and demographic characteristics from the participant and criminal/judicial process, see Hypotheses 2a and 2b on page 8.

We have now added the following text on page 16:

"Background characteristics (i.e., gender, age, and educational level, kinship to the deceased, and time since loss) and accident-related stressors (i.e., single vs. multiple loss, witnessed the accident, and status of legal trial) will be assessed with single items."

Please see also top of page 18 for details:

"To achieve research aim 2b, multiple regression analyses will be used to examine to what extent residual gain scores of PCBD, PTSD, and depression varies as function of a) background characteristics, including gender (male/female), age (in years), and educational level (low/high), kinship to the deceased (child/spouse vs other), and time since loss (in years) and b) accident-related stressors, including number of losses (single vs multiple), witnessing the accident (yes/no), and status of legal trial (not applicable/on-going/completed)."

While interesting, we did not include media intrusion to limit the length of the surveys.

4. It would be also important to clarify:

- any other current or past Psychology interventions
- and/or medical/pharmacological treatment

Response: We now clarified how we plan to assess past psychological support on page 17:

"We will also include two dichotomous items (yes/no) at T1 to assess psychological support received prior to participation in the study, namely: "Did you ever receive support from a psychologist, therapist or psychiatrist, for your own emotional/mental problems, prior to the loss of your loved one due to a traffic accident?" and "Did you ever receive support from a psychologist, therapist or psychiatrist, for your own emotional/mental problems, related to the loss of your loved one due to a traffic accident?"

In the Netherlands, medical/pharmacological treatments are usually prescribed by psychiatrists, so we believe that these two items adequately assess psychological and pharmacological treatment.

5. Pg 11 - Can you clarify if the research team is going to be (or not) involved in psychology provision?

Response: Two members of the research team (Paul Boelen and Jos de Keijser), who are both registered clinical psychologists, are available for provision of the treatment.

We have now added this information on page 9:

"In total six therapists (including authors PB and JdK who are registered clinical psychologists) will guide the participants online; participants will receive feedback from the same therapist each time."

6. Pg 12 - Can you provide more detail about the clinical judgment clinicians will have to make in order to following your inclusion and exclusion criteria. Is this going to be discussed and agreed to reduce bias? If so, how?

Response: Since we removed the face-to-face arm of the RCT due to COVID-19 all contact between participants and therapists will be online. Clinical judgment for screening of exclusion criteria (e.g., substance disorder) has now been removed from the protocol. The paragraph on page 11 now reads:

“To be eligible for study participation, the person must 1) be a family member, spouse, or friend of a person who died due to a traffic accident at least one year earlier, 2) be ≥18 years of age, and 3) meet DSM-5 criteria for PCBD and/or PTSD and/or experience clinically relevant depression, based on questionnaire scores (see below for more details). People are excluded when they do not master the Dutch language or have no Internet access.”

7. Pg - Can you clarify what training and supervision therapists will receive?

Response: We have now provided more information about the training and supervision on page 9 and 10:

“The therapists will receive a training, provided by LL, PB, and JdK, on the use of the treatment protocol of this intervention study. In preparation for the training, therapists read all treatment materials and a selection of grief treatment literature. Instructions about the use of the online treatment interface will be given by its developers. During a 5-hour face-to-face group meeting the rationale of the online treatment will be explained and research procedures will be discussed. In a 2-hour online video-meeting outstanding questions regarding the treatment and the research project will be answered. Supervision (by telephone or mail) by PB and JdK is possible on request, for instance when therapists encounter difficulties in treatment.”

8. Will trauma-informed CBT be considered?

Thank you for this suggestion. We choose to evaluate the effects of online grief-specific CBT, and not trauma-specific treatment, because prior research shows that grief-specific treatment yields stronger effects than treatments focused on other forms of mental health problems, see page 6:

“Grief-specific CBT has been demonstrated to be the most effective treatment for bereaved people with elevated grief levels(21–24).”

9. What safeguarding arrangements do you have in place?

Response: We are not completely sure what the reviewer means with “safeguarding arrangements”, but as mentioned on page 19:

“This study has been approved by a local ethics committee (METc UMCG: ID number: M20.252121). The study will be conducted according to the principles of the Declaration of Helsinki (8th version, 2013) and in accordance with the Medical Research Involving Human Subjects Act.”

We therefore follow national and international guidelines regarding clinical research with humans.

10: Can you clarify if sessions will be 1:1 or group setting?

Response: Thank you pointing this out. We have now added this information on page 12:

“Online CBT will consist of eight one-on-one sessions, called lessons, offered within a timeframe of 12 weeks.”

Overall, manuscript reads well, however some paragraphs are too lengthy and lack of clarity. Thank you.

Response: We removed parts of the text as a result of removing the face-to-face treatment arm (as explained at the beginning of this letter), but we also carefully checked and removed text throughout the manuscript to reduce the length of paragraphs. Based on the reviewers' comments we provided more details on certain aspects of our study to improve clarity.

Reviewer: 2

Reviewer Name
KEE-HONG CHOI

Institution and Country
KOREA UNIVERSITY, KOREA

Please state any competing interests or state 'None declared':
NONE

Please leave your comments for the authors below

1. The current study protocol proposed to examine the efficacy of face-to-face CBT and online CBT as compared to waitlist control group in reductions of PCBD, PTSD, and depressive symptoms. In addition, the authors proposed to explore potential moderators and mediators of change in those symptoms.

The authors proposed two important but distinct study aims: (1) efficacy of face-to-face and online CBT, and (2) treatment mechanisms of CBT. Even though two study aims are important and interesting, the mixture of the two aims would cause concerns especially about sample size. The authors said that due to a small sample, the online CBT is not compared to face-to-face CBT even though online CBT for PCBD is quite novel (and having distinct treatment components as indicated by the authors) and unknown for its efficacy. In addition, since investigation of treatment mechanisms requires a large number of subjects, it is unclear why the authors include the online CBT (without examining its efficacy) in the study design for exploring treatment mechanisms. For the treatment mechanisms, it would be better increasing sample size of the face-to-face CBT.

Response: Thank you, Kee-Hong Choi for your helpful feedback on our manuscript. We had to make changes to our study protocol due to the COVID-19 outbreak, explained on page 1 of this response letter. This resulted in removing the face-to-face arm and consequently we rephrased the first aim of our study, which now reads (page 7):

“Our first aim is to examine the effectiveness of online CBT (vs. a waiting list control condition) in reducing symptom levels of PCBD, PTSD, and depression in people bereaved by a traffic accident.”

Regarding the treatment mechanisms, we agree that the anticipated sample size may be too small to conduct mediation analyses. So instead, we now will examine possible correlates of change by calculating residual gain scores. Wording referring to mediators/moderators are replaced by wording referring to “correlates of change”. See for instance page 8:

“Our second aim is to explore correlates of change. Based on prior research and theories(12,16,19), we expect that reductions in negative cognitions, avoidance behaviors, state anger, a sense of unrealness, and improvement in self-efficacy are related to reductions in PCBD, PTSD, and depression levels in online CBT (Hypothesis 2a). Additionally, we aim to explore whether background characteristics (i.e., gender, age, and educational level, kinship to the deceased, and time since loss) and accident-related stressors (i.e., single vs. multiple loss, witnessing the accident, and status of legal trial) are related to treatment effects (Hypothesis 2b). We have no specific expectations regarding these associations because prior treatment studies in bereaved people showed inconsistent results(24,25,40). However, based on clinical experience, we expect that accident-related stressors are associated with treatment effects, such that multiple loss, witnessing the accident, and on-going legal trial negatively impact treatment effects.”

On page 18 we described the statistical method of this approach in more detail.

2. It would be helpful for the readers if the authors commented on the effects of moderators in other related studies (though with a different population), such as the fact that male, younger age, and higher level of education were associated with outcomes in PTSD studies (Ehlers et al., 2013). Moreover, I encourage the authors to address their hypothesis regarding the background characteristics based on clinical experience AND the results from other related studies. In the manuscript, only the hypothesis concerning accident-related stressors was noted.

Response: Thank you for raising this point. On page 8 we stated: *“We have no specific expectations regarding these associations because prior treatment studies in bereaved people showed inconsistent results(24,25,40).”* We did not include hypotheses based on PTSD studies in trauma samples, because the few studies on these possible moderators in grief treatment studies are inconclusive.

3. How to control for the practice effects on repeated measures of waitlist controls after allocating to either face to face or online CBT? What if participants who are allocated to waitlist control become no longer eligible for the study due to their natural recovery after 20 weeks?

Response: We include people in our RCT based on their PCBD, PTSD, and depression levels at baseline. In case people recover during the waiting period and no longer meet diagnostic criteria for

PCBD, PTSD, and elevated depression they are still able to start the online CBT after the waiting period of 20 weeks. We will report on this and we have now added information on this issue on page 18:

“Furthermore, percentages of people meeting diagnostic criteria for PCBD, PTSD, and depression will be calculated for each measurement occasion...”

4. Please elaborate on the qualifications of the personalized therapists for online sessions. For example, are they licensed and registered psychologists who will also receive trainings that are qualitatively equivalent to the 8-hour training that the face-face CBT therapists would receive? (it was somewhat unclear in the manuscript).

Response: We now provided more information about this (see also our response to comment 7 of Reviewer 1) on page 9.

5. Please provide more information on the video-therapist that will conduct the online lessons. Who will be the person conducting the online-lessons (e.g., licensed psychologist, gender, age, etc)? Will the instructor of the online-lessons be the same as the one who will be providing weekly asynchronous feedbacks? This seems like an important issue as the third hypothesis of the study examines the extent to which therapeutic alliance differ in each condition.

Response: Thank you for pointing this out. We now clarified this by providing more details about the video-therapist and the online therapist on page 13:

“All information and assignments are presented in an online framework, offered via a secure website. Participants receive online written information that consists of psychoeducation, information about treatment content and structure, and homework assignments. As part of the online treatment, participants also listen to a video-therapist verbally sharing parts of information that are also presented in text. The video-therapists are two therapists from the Traumatic Loss Network; one male and one female and both middle-aged. At the start of the treatment the video-therapists introduce themselves and the participant is asked to select one of the video-therapists. The information shared by these video-therapists are recorded in video-messages in which they read parts of the texts out loud. Each participant therefore receives the same information from a video-therapist. Direct contact with the video-therapist is not possible. Participants receive weekly asynchronous written feedback from one online therapist on each assignment that they complete online. As mentioned earlier, six online therapists are trained to guide the participants.”

Because the participants are guided by one online therapist (and only listen to recorded video-messages by video-therapists) the therapeutic alliance measure is related to the online therapist that guide them during the treatment and provide feedback on the assignments.

6. Will there be a specific guideline for providing weekly asynchronous feedbacks? For instance, will the therapists be encouraged to provide feedbacks within a certain time period such as 2 or 3 days?

Response: We now provided details on this at the top of page 13:

“The online therapists are instructed to contact the participant twice a week; once to encourage participants to log in and complete assignments and once to provide feedback on assignments. In total, they spend 30 minutes per week on reading assignments and providing feedback.”

Who will administer measures, and how to confirm that the measures are administered blind to a study design? Would therapists be blind to a study design?

Response: A member of the research team will administer the measures to the participants. We added this information on page 9: *“A link to online questionnaires will be sent to the participants by a non-blinded member of the research team at each time-point.”*

More detailed information about the differences in therapeutic contact (e.g., different doses of guidance, different forms of communication mode) between two therapeutic arms should be helpful.

Response: Because we removed the face-to-face CBT arm in our study, this is not applicable anymore.

VERSION 2 – REVIEW

REVIEWER	Kee-Hong Choi Korea University, Korea
REVIEW RETURNED	Korea University, Korea 05-Jul-2020

GENERAL COMMENTS	The revised manuscript has improved its clarity about the study design and procedures. Minor points are presented for further clarifications: Please specify video- and online-therapists' expertise (specialist vs nonspecialist) just as face-to-face therapists. It is concerned about the validity of the therapeutic alliance measure, as the participants will be receiving instructions from two separate therapists according to your study design (e.g., information from video therapist and feedback from another therapist). Although the study will be carried out under the assumption that the therapeutic alliance measure is related to the online therapist that provides feedback on the assignments, we are concerned that having two distinct therapists in the treatment will make it difficult to determine to which the measure is related. If you're unable to address this issue due to the limitation of the study, we suggest that you include it as a part of the limitation in the discussion section.
---

VERSION 2 – AUTHOR RESPONSE

Reviewer(s)' Comments to Author:

Reviewer: 2
Reviewer Name
Kee-Hong Choi

Institution and Country
Korea University, Korea

Please state any competing interests or state 'None declared':
None

Comment: Please leave your comments for the authors below
The revised manuscript has improved its clarity about the study design and procedures. Minor points are presented for further clarifications:

Response: Thank you very much for your positive words.

Comment: Please specify video- and online-therapists' expertise (specialist vs nonspecialist) just as face-to-face therapists.

Response: On page 9 we now specified the expertise of the online therapists:

"In line with prior treatment studies from our research group(40,41), the online treatment is guided by governmentally licensed psychologists, connected with a Dutch informal "traumatic loss network" of therapists specialized in treating emotional distress following traumatic loss."

On page 13 we now explained:

“The video-therapists are two members from the traumatic loss network; one male and one female psychotherapist who are middle-aged and specialized in treating bereaved people.”

Comment: It is concerned about the validity of the therapeutic alliance measure, as the participants will be receiving instructions from two separate therapists according to your study design (e.g., information from video therapist and feedback from another therapist). Although the study will be carried out under the assumption that the therapeutic alliance measure is related to the online therapist that provides feedback on the assignments, we are concerned that having two distinct therapists in the treatment will make it difficult to determine to which the measure is related. If you're unable to address this issue due to the limitation of the study, we suggest that you include it as a part of the limitation in the discussion section.

Response: We have added this as limitation on page 22:

“In addition, participants might experience difficulties with completing the mid-treatment assessment of the therapeutic relationship because the video-therapist that provides information through recorded video messages (interaction between video-therapist and participant is not possible) might be a different person than the online therapist who provides personal written feedback twice a week. Although the instructions of the therapeutic alliance measure explicitly refer to the interaction with the online therapist (not the video-therapist), this might still be confusing for some participants.”